# Inhibition of the oligosaccharyl transferase in *Caenorhabditis elegans* that compromises ER proteostasis suppresses p38-dependent protection against pathogenic bacteria

**Dae-Eun Jeong**[1☯¤a], **Yujin Lee**[2☯], **Seokjin Ham**[2☯], **Dongyeop Lee**[1¤b], **Sujeong Kwon**[2], **Hae-Eun H. Park**[2], **Sun-Young Hwang**[1¤c], **Joo-Yeon Yoo**[1], **Tae-Young Roh**[1], **Seung-Jae V. Lee**[2]*

**1** Department of Life Sciences, Pohang University of Science and Technology, Pohang, Gyeongbuk, South Korea, **2** Department of Biological Sciences, Korea Advanced Institute of Science and Technology, Yuseong-gu, Daejeon, South Korea

☯ These authors contributed equally to this work.
¤a Current address: Department of Pathology, Stanford University School of Medicine, Stanford, California, United States of America
¤b Current address: Department of Biology, Howard Hughes Medical Institute, Massachusetts Institute of Technology, Cambridge, Massachusetts, United States of America
¤c Current address: Program in Molecular, Cell and Cancer Biology, University of Massachusetts Medical School, Worcester, Massachusetts, United States of America
* seungjaevlee@kaist.ac.kr

**Data Availability Statement:** RNA seq. raw data and processed data are available in Gene Expression Omnibus (https://www.ncbi.nlm.nih.

## Abstract

The oligosaccharyl transferase (OST) protein complex mediates the N-linked glycosylation of substrate proteins in the endoplasmic reticulum (ER), which regulates stability, activity, and localization of its substrates. Although many OST substrate proteins have been identified, the physiological role of the OST complex remains incompletely understood. Here we show that the OST complex in *C. elegans* is crucial for ER protein homeostasis and defense against infection with pathogenic bacteria *Pseudomonas aeruginosa* (PA14), via immune-regulatory PMK-1/p38 MAP kinase. We found that genetic inhibition of the OST complex impaired protein processing in the ER, which in turn up-regulated ER unfolded protein response (UPR$^{ER}$). We identified vitellogenin VIT-6 as an OST-dependent glycosylated protein, critical for maintaining survival on PA14. We also showed that the OST complex was required for up-regulation of PMK-1 signaling upon infection with PA14. Our study demonstrates that an evolutionarily conserved OST complex, crucial for ER homeostasis, regulates host defense mechanisms against pathogenic bacteria.

## Author summary

N-linked glycosylation is essential for the function of various proteins, but its effects on physiology at an organism level remain poorly understood. Using the roundworm *Caenorhabditis elegans*, we show that the oligosaccharyl transferase (OST) complex, which mediates the N-glycosylation of substrate proteins in the ER, reduces susceptibility to

gov/geo, GSE134687). All other data are within the manuscript and Supporting Information files.

**Funding:** This research was supported by grants NRF-2016R1E1A1A01941152 and NRF-2017R1A5A1015366 funded by the Ministry of Education, Science and Technology through the National Research Foundation (NRF) of Korea to S-JVL. The funders had no role in study design, data collection and analysis, decision to publish, or preparation of the manuscript.

**Competing interests:** The authors have declared that no competing interests exist.

pathogenic bacteria, *Pseudomonas aeruginosa*. We find that OST enhances defense against *P. aeruginosa* via maintenance of ER unfolded protein response (UPR$^{ER}$) and up-regulation of cytosolic p38 MAP kinase signaling. Our findings propose an intriguing model for the organellar crosstalk between the ER and the cytosol in host defense mechanisms. Because the OST complex components are highly conserved among eukaryotes, our study on the regulation of cellular signaling and *C. elegans* physiology by the OST complex will provide an insight into the function of its mammalian counterpart.

## Introduction

N-linked protein glycosylation is an abundant posttranslational modification that regulates protein stability, localization, and activity [1–3]. Defects in the protein N-glycosylation process lead to various pathophysiological consequences, such as abnormal development and metabolic disorders [4]. Protein N-glycosylation is also implicated in immunity. For example, mutations in genes that act in the N-glycosylation pathway cause defects in both the innate and adaptive immune systems [5–7]. Nevertheless, the contribution of protein N-glycosylation to immunity at the organism level remains incompletely understood.

Protein N-glycosylation is mediated by enzymatic processes that transfer mono- or polysaccharides to asparagine residues in a consensus motif (N-X-S/T; X stands for any amino acid except proline) of substrate proteins [8]. At the ER membrane, the oligosaccharyl transferase (OST) complex mediates N-linked glycosylation by transferring the glycans from glycan precursors to asparagine residues in peptides [9, 10]. The eukaryotic OST complex consists of highly conserved subunits, ribophorin I, ribophorin II, OST48, OST4, STT3 (staurosporine and temperature sensitive 3), N33/Tusc3 (tumor suppressor candidate 3) or IAP (implantation-associated protein), and DAD1 (defender against apoptotic death 1) [9, 11]. Although various N-glycosylated proteins have been identified in many metazoans such as the roundworm *Caenorhabditis elegans* [12, 13], the physiological roles of the OST complex remain largely unexplored.

*C. elegans* is an excellent model to study genetic mechanisms underlying the regulation of anti-bacterial host immunity [14–19]. In nature, *C. elegans* live in soil and/or rotten fruits rich in microbes, and contain many pathogenic bacteria in the gut [20, 21]. Various genetic factors that regulate the immune responses of *C. elegans* against pathogenic bacteria, including *Pseudomonas aeruginosa* (PA14) have been identified [18]. In particular, PMK-1/p38 MAP kinase is activated upon infection with PA14, leading to up-regulation of transcription factors, including ATF-7, ELT-2, and SKN-1 [22–28]. In addition, ER unfolded protein response (UPR$^{ER}$) regulators, including IRE-1 (a transmembrane serine/threonine protein kinase) and XBP-1 transcription factor, are critical for development under PA14 infection and protection of adult worms against PA14 [29–31].

In the current study, we show that the OST complex that mediates protein N-linked glycosylation is critical for the innate immunity of *C. elegans* against PA14 infection. Genetic inhibition of the OST complex caused ER stress, possibly through the accumulation of unprocessed proteins in the ER, which in turn increased UPR$^{ER}$. In addition, OST complex was required for the induction of various PMK-1/p38 MAP kinase-regulated genes under PA14-infected conditions. We also identified vitellogenin-6 (VIT-6) as an OST-dependent glycosylated protein, crucial for protection against PA14. Furthermore, we showed that inhibiting the OST complex substantially increased susceptibility to PA14 through down-regulating PMK-1. Our study suggests that the OST complex maintains ER protein homeostasis and enhances defense against pathogenic bacteria through activation of PMK-1 signaling.

## Results

### The oligosaccharyl transferase (OST) complex is required for the maintenance of *C. elegans* survival upon infection with *Pseudomonas aeruginosa*

Our previous RNAi screen for anti-bacterial immunity-modulatory genes encoding mitochondrial factors [32] tested many non-mitochondrial genes which were not included in the paper due to lack of relevance (S1 Table). Among them, RNAi targeting ZK686.3/N33/Tusc3, an OST complex component, substantially decreased the survival of *C. elegans* to PA14 (Fig 1A, slow killing, small lawn where worms effectively avoid pathogenic PA14). We then found that RNAi targeting each of four other OST complex components, *ribo-1*/ribophorin I, *stt-3*/STT3, *ostb-1*/OST48 and *ostd-1*/ribophorin II [11], substantially reduced worm survival under PA14 infection (Fig 1A, S1A and S1B Fig). Furthermore, a reduction-of-function mutation of *stt-3*, which altered tyrosine 550 to alanine in a catalytic subunit of OST ([33], see Fig 1B legend and S1C Fig) by using CRISPR/Cas9-based genome editing drastically increased susceptibility to PA14 (Fig 1B). We found that RNAi targeting each of the OST components enhanced the accumulation of GFP-labeled PA14 in the intestinal lumen (Fig 1C and 1D). Additionally, RNAi targeting *stt-3* reduced worm survival on the big lawn of PA14 where worms do not have space for an avoidance behavior (Fig 1E). In addition to slow killing conditions, in which PA14 kill hosts by colonization at the intestinal lumen, in fast killing conditions PA14 secretes toxic factors that compromise host health [34–36]. We found that genetic inhibition of *stt-3* increased sensitivity to PA14 under fast killing conditions as well (Fig 1F). These data suggest that OST components are required for the protection of *C. elegans* against pathogenic PA14 and that reduced survival by inhibition of OST is not due to defects in PA14 avoidance behavior. In contrast, RNAi targeting each OST component had small effects on the lifespan of *C. elegans* fed with standard *E. coli* foods (Fig 1G; *ribo-1* RNAi data are consistent with [37]), or survival upon infection with pathogenic *E. coli* (Fig 1H) or *E. faecalis* (Fig 1I). These data indicate that enhanced sensitivity of worms to PA14 by genetic inhibition of OST complex is not simply caused by potential lethality or sickness. Together, the OST complex appears to specifically modulate susceptibility to PA14 infection rather than generally affecting worm survival.

### The OST complex regulates the proper localization of vitellogenin-6, crucial for survival on PA14

We then sought to identify OST substrates that were crucial for immunity against PA14, using a small-scale two-dimensional polyacrylamide gel electrophoresis (2D-PAGE) followed by glycoprotein staining and peptide mass fingerprinting (See Materials and Methods for details). Vitellogenin-6 (VIT-6) was one of three recovered *stt-3*-dependent N-glycosylated proteins (S2 Table), consistent with a previous N-glycoproteomic study in *C. elegans* [12]. *vit-6* is expressed in the intestinal cells of *C. elegans* ([38], WormBase database), and subsequently secreted and imported into the eggs. We showed that *stt-3* RNAi impaired the strong localization of VIT-6::mCherry in the eggs (Fig 2A). These data suggest that N-glycosylation of VIT-6 is critical for its proper secretion, and that VIT-6 is an OST substrate that affects physiological responses. A previous report showed that mutations that cause abnormal accumulation of vitellogenins, including VIT-6, impair protection against PA14 [39]. Here we found that RNAi knockdown of *vit-6* substantially decreased the survival of wild-type worms on PA14 (Fig 2B), without affecting normal lifespan on an *E. coli* diet (Fig 2C). In addition, *vit-6* RNAi and *stt-3* RNAi did not have an additive effect on the susceptibility of worms to PA14 (Fig 2D). Although genetic interaction analysis with double RNAi knockdown has limitations for

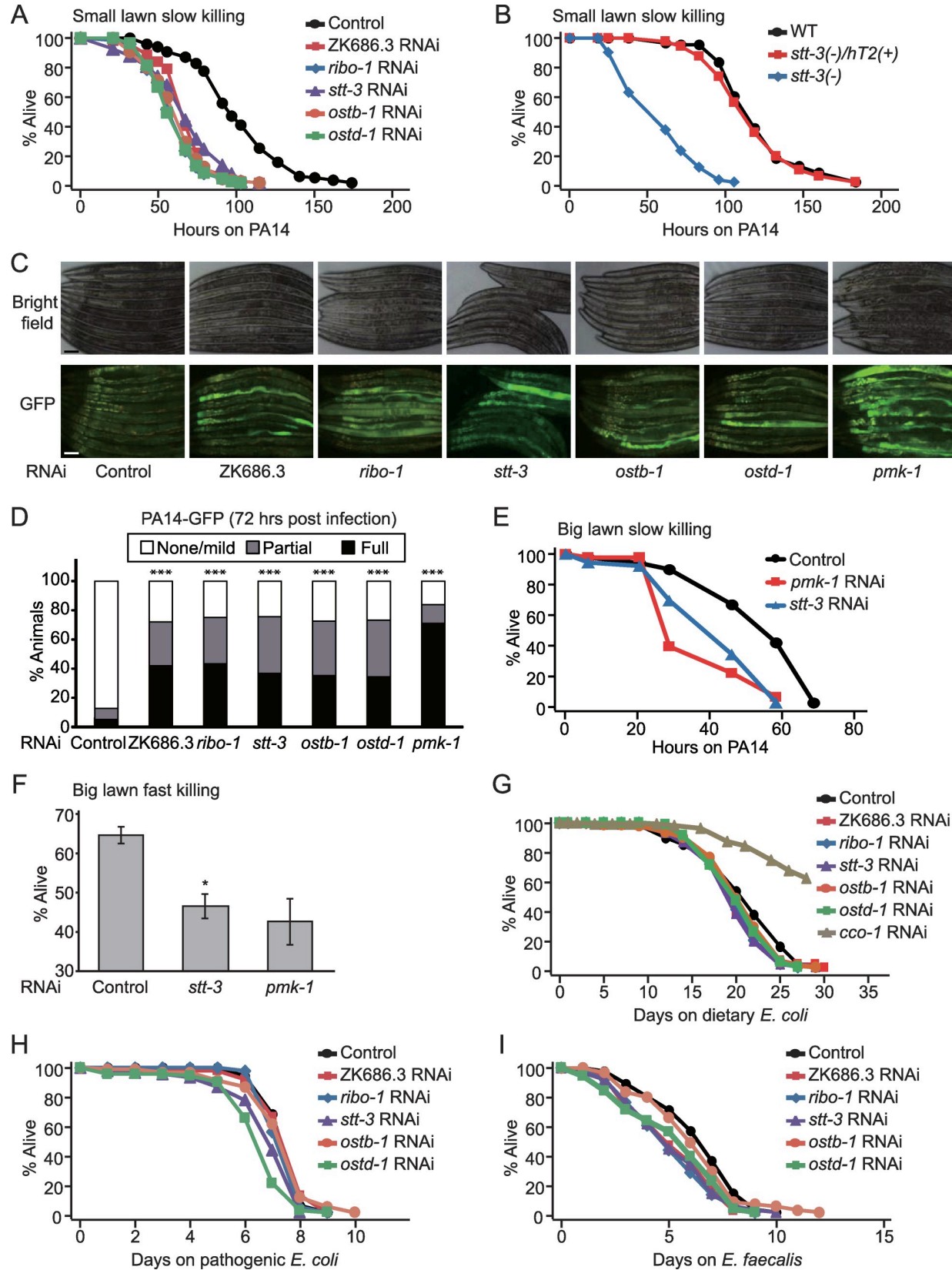

**Fig 1.** *C. elegans* **OST complex is required for protection against and clearance of PA14.** (**A-B**) Genetic inhibition of OST complex by RNAi targeting each component (ZK686.3, *ribo-1*, *stt-3*, *ostb-1*, or *ostd-1*) (**A**) or by a mutation in *stt-3(syb458)* [*stt-3(-)*] (**B**) reduced the survival of *C. elegans* under PA14-infected conditions. WT: wild-type (N2), *stt-3(-)/hT2(+)*: a balanced heterozygote. We replaced 550th tyrosine residue in STT-3 with alanine for generating a reduction of function allele (S1C Fig), because the orthologous change (Y519A) in yeast STT3 caused mild growth defects without causing lethality [33]. *stt-3(syb458)* homozygotes were able to develop to L4/young adult stage but were sterile. These phenotypic observations suggest that *stt-3(syb458)* is a reduction of function allele. (**C**) Shown are representative images of worms that were pre-treated with control RNAi and RNAi targeting each indicated OST component, after PA14-GFP exposure for 72 hrs. *pmk-1* RNAi was used as a positive control. Scale bar indicates 100 μm. (**D**) Semi-quantification of GFP-labeled PA14 levels in the intestinal lumen of worms in panel **C** (n ≥ 20 from two independent experiments). *p* values were calculated by using Chi-squared test (***$p < 0.001$). We also found that *stt-3* RNAi slightly reduced pumping rates of worms (S1D Fig). Thus, the increased accumulation of PA14 by inhibition of OST seems unlikely due to increased PA14 uptake. (**E-F**) Inhibition of OST complex by *stt-3* RNAi reduced the survival of worms both under slow (**E**) and fast killing (**F**) conditions of which the whole surface of plates was covered by PA14 (big lawn of PA14). (**G**) Perturbation of OST complex by RNAi targeting each of OST components had small effects on lifespan fed with normal dietary *E. coli* diets. RNAi targeting *cco-1* (cytochrome c oxidase-1 subunit) that increases lifespan [83] was used as a positive control. (**H**) Knockdown of each of three components (ZK686.3, *ribo-1*, and *ostb-1*) of the OST complex did not alter the survival of worms upon infection with pathogenic *E. coli*. RNAi targeting *stt-3* or *ostd-1* slightly reduced the survival on pathogenic *E. coli*. (**I**) The survival time of worms treated with each of four OST components (ZK686.3, *ribo-1*, *stt-3*, and *ostd-1*) was variable under *E. faecalis*-infected conditions. See S3 Table for additional repeats and statistical analysis for survival assay results in this figure.

interpretation, this result is consistent with the possibility that VIT-6 and OST regulate the survival of animals on PA14 by acting in the same genetic pathway. Together, these results suggest that the OST complex promotes the adequate N-glycosylation of proteins such as VIT-6, which contributes to the protection of worms against PA14.

## Inhibition of the OST complex induces UPR$^{ER}$

In *C. elegans*, PA14 infection induces substantial transcriptional changes in immunity-regulating genes [23, 24]. Because the OST complex was required for immunity against PA14, we asked whether the OST complex altered transcriptional responses upon PA14 infection. We specifically determined the transcriptomic changes caused by knockdown of *stt-3* with or without PA14 infection. By performing mRNA sequencing (RNA-seq) analysis, we first noticed that *stt-3* RNAi substantially altered transcriptome profile on normal *E. coli* diets without PA14 infection (Fig 3A). Notably, gene ontology (GO) terms related to UPR$^{ER}$ as well as immune responses were over-represented among the 208 genes up-regulated by *stt-3* RNAi (Fig 3B, S4 Table, fold change > 2, Benjamini and Hochberg (BH)-adjusted *p* value < 0.05 with 17,430 genes), whereas 128 down-regulated genes were enriched in GO terms of metabolic processes (S2A Fig, S4 Table). Consistently, *stt-3* RNAi increased the levels of *hsp-3*, *hsp-4*, and *enpl-1*, which encode major ER chaperones induced by UPR$^{ER}$ (Fig 3A). We confirmed the induction of *hsp-4* by *stt-3* RNAi using an *hsp-4p::GFP* reporter (Fig 3C and 3D) and qRT-PCR (Fig 3E). The induction of *hsp-4p::GFP* by *stt-3* RNAi was completely suppressed by mutations in *xbp-1* or *ire-1*, which encode key UPR$^{ER}$ regulators (Fig 3C and 3D). We obtained consistent qRT-PCR results for the mRNA levels of *hsp-3* and *hsp-4* (S2C and S2D Fig). In addition, the ratio of spliced (active) to unspliced (inactive) *xbp-1* mRNA forms was increased by *stt-3* RNAi (Fig 3F–3H and S2E Fig). These results indicate that inhibition of the OST induces UPR$^{ER}$ through the IRE-1/XBP-1 signaling axis.

## The OST complex contributes to PMK-1-dependent transcriptional changes upon infection with PA14

We then analyzed the role of the OST complex in transcriptomic changes caused by infection with PA14 (Fig 4A and S3A Fig). In animals infected with PA14, 1,440 genes were up-regulated and 2,309 genes were down-regulated (Fig 4A, fold change > 2, BH-adjusted *p* value < 0.05 with 17,430 genes), indicating a broad transcriptional remodeling. We noticed that *stt-3* RNAi reduced (fold change > 2) the expression of 569 genes up-regulated by PA14 (Fig 4A

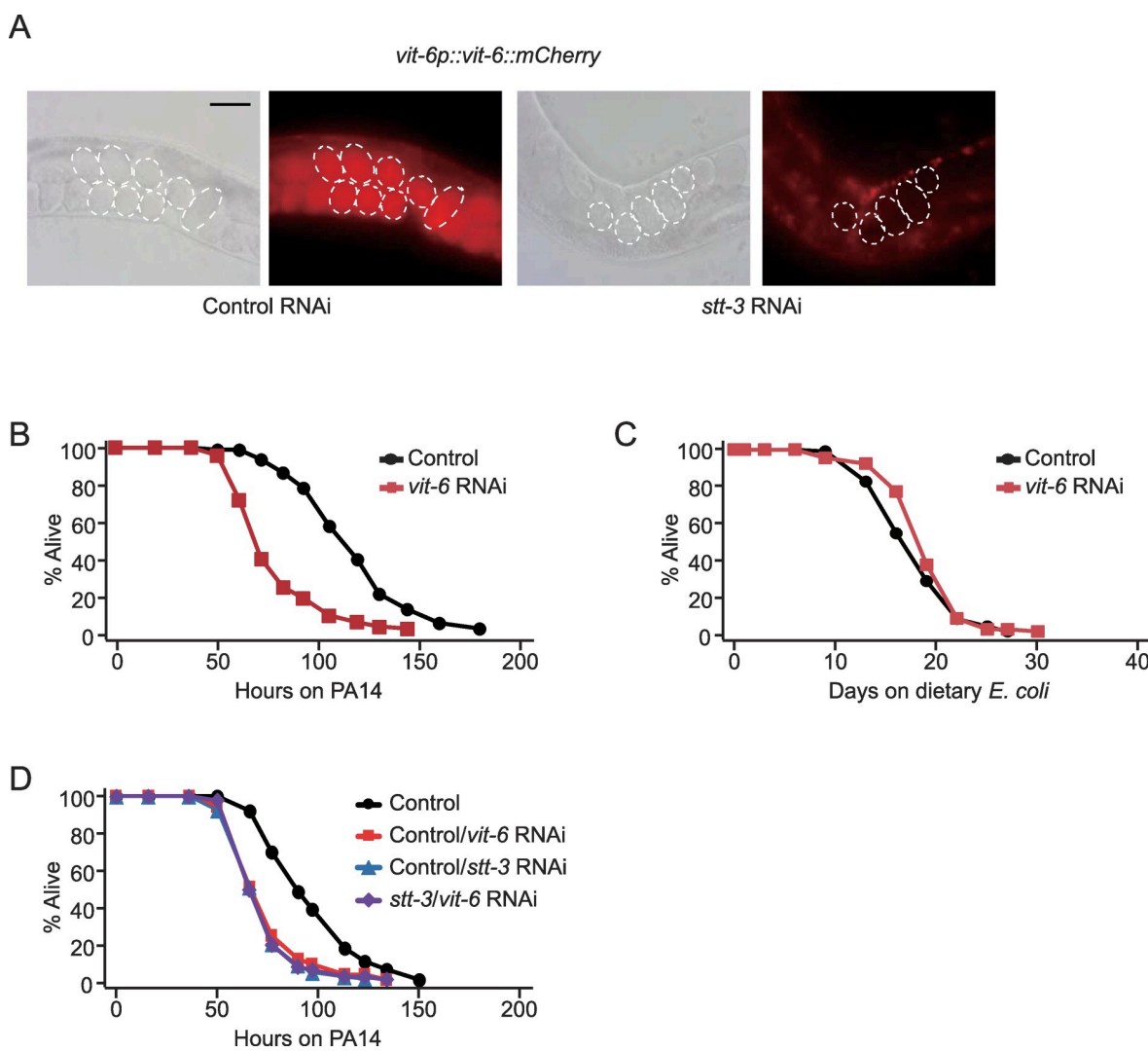

**Fig 2. PA14 infection promotes OST-dependent N-glycosylation of VIT-6 that is responsible for ER homeostasis.** (**A**) Fluorescence images of *vit-6::mCherry* transgenic animals treated with control or *stt-3* RNAi. Dashed lines indicate embryos in the day 2 adult worms. Scale bar indicates 100 μm. (**B-C**) RNAi targeting *vit-6* reduced the survival of wild-type (N2) worms on PA14 (**B**) without decreasing lifespan on *E. coli* diets (**C**). (**D**) RNAi targeting *vit-6* did not affect the survival of worms treated with *stt-3* RNAi. See S3 Table for additional repeats and statistical analysis for survival assay results in this figure.

and S5 Table), among which immune defense-associated GO terms were substantially enriched (Fig 4B). In contrast, 569 *stt-3*-dependent PA14-repressed genes were not enriched with immune responses (S3B Fig and S5 Table). We then examined which genetic pathways acted together with the OST complex for the defense against PA14, by comparing our RNA-seq data with previously published *C. elegans* transcriptome data (Fig 4C) [24, 28, 40–48]. We found that genes whose induction is dependent on ELT-2, PMK-1, and ATF-7 were substantially up-regulated by PA14 in an *stt-3*-dependent manner (Fig 4D and 4E). These data are consistent with the previous report showing that ELT-2 and ATF-7 cooperate to induce a subset of PMK-1-regulated immune genes [27]. In addition, genes whose expression was increased by the inhibition of *vhp-1*/MAP kinase phosphatase, which leads to PMK-1 activation [49], were significantly enriched with genes up-regulated by PA14 in an *stt-3*-dependent

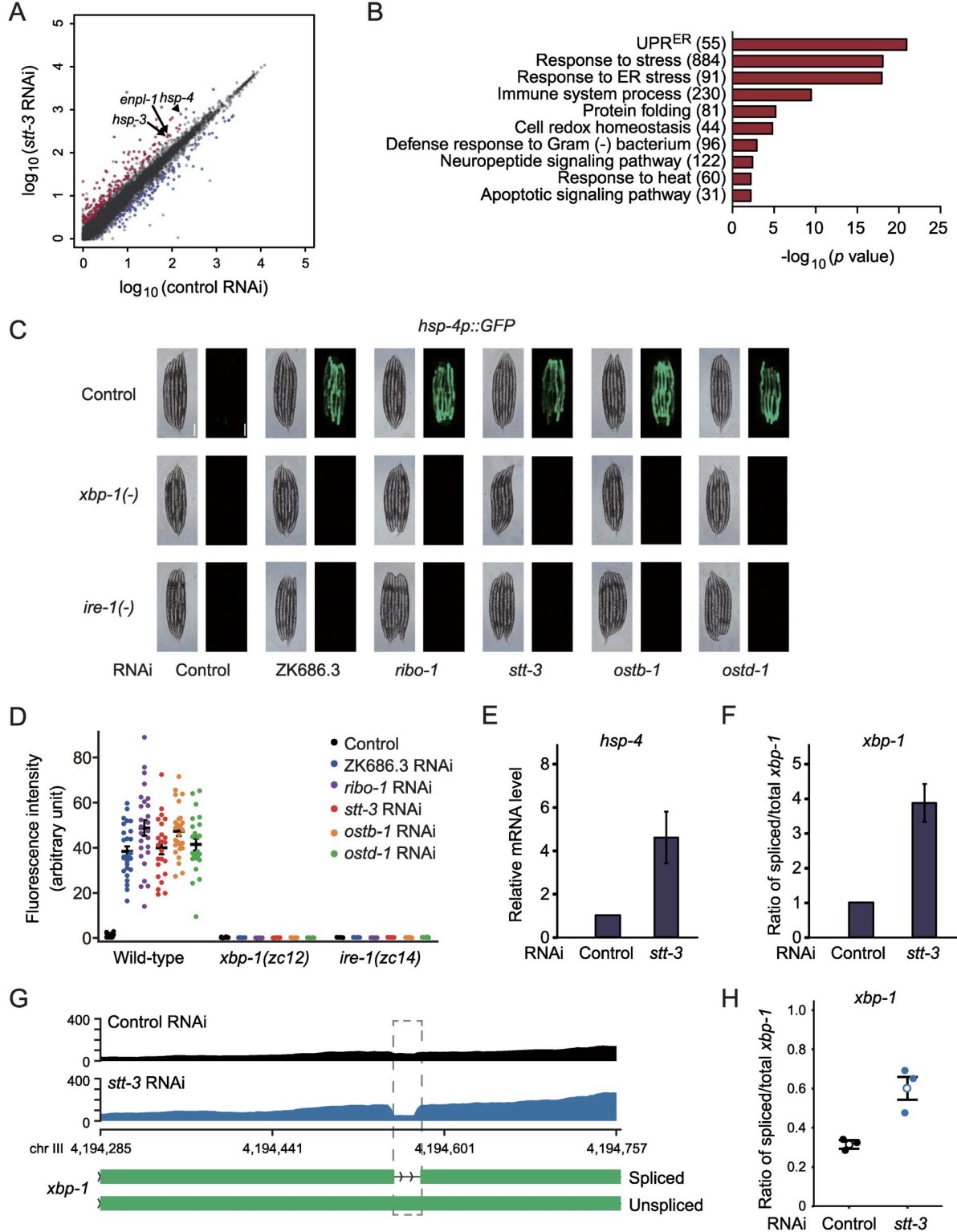

**Fig 3. OST complex maintains ER protein homeostasis.** (**A**) A scatter plot showing the effects of RNAi targeting *stt-3* on the expression levels of mRNAs. Red dots and blue dots indicate 208 up- and 128 down-regulated genes, respectively (fold change > 2, Benjamini and Hochberg (BH)-adjusted *p* value < 0.05). (**B**) Overrepresented GO terms of genes induced by *stt-3* RNAi. UPR[ER]: ER unfolded protein response. The numbers of genes for individual terms are indicated in parentheses. (**C**) RNAi targeting each OST component (ZK686.3, *ribo-1*, *stt-3*, *ostb-1*, or *ostd-1*) increased the expression of *hsp-4p::GFP* in wild-type animals but not in *xbp-1(zc12)* [*xbp-1(-)*] or in *ire-1(zc14)*

[*ire-1(-)*] mutants. Scale bar indicates 250 μm. (**D**) Quantification of data in panel **C** (n ≥ 24 from two independent experiments). (**E-F**) qRT-PCR analysis data showing mRNA levels of *hsp-4* (**E**) and the ratio of spliced/total *xbp-1* mRNAs (**F**) in worms treated with control or *stt-3* RNAi (n ≥ 3). Error bars indicate SEM. *p* values were calculated by two-tailed Student's *t*-test (*$p < 0.05$, **$p < 0.01$, ***$p < 0.001$). (**G**) An intron retention of *xbp-1* transcripts in a representative RNA-seq sample. A grey box with a dotted line indicates the location of alternative splicing. (**H**) The ratio of spliced/total *xbp-1* mRNAs in triplicate RNA-seq samples. Error bars indicate SEM.

manner (Fig 4D). We found that *stt-3*-independent PA14-induced genes had substantially smaller overlaps with genes up-regulated by SEK-1 (PMK-1 upstream kinase), PMK-1 or ELT-2, or by inhibition of VHP-1 than *stt-3*-dependent PA14-induced genes did (Fig 4F). In contrast, genes whose expression levels are changed by the genetic modulation of each of the other immune-regulating transcription factors, ZIP-2/bZIP [45], SKN-1/NRF2 [26], HSF-1/heat shock factor 1 [50] and DAF-16/FOXO [24], were independent of *stt-3* (S4 Fig). Our analysis of chromatin-immunoprecipitation sequencing (ChIP-seq) data [51] also suggests the binding of ELT-2, a GATA transcription factor, to the promoters and bodies of *stt-3*-dependent PA14-induced genes (Fig 4G). Moreover, a GATA transcription factor-binding motif (TTATCA) was predominantly located in the promoter regions (-1 kb to +100 bp) of *stt-3*-dependent PA14-induced genes (Fig 4H). These results suggest that transcriptional responses triggered by the PMK-1/p38 MAP kinase signaling upon PA14 infection require the functional OST complex.

Next, we confirmed our RNA-seq. data by showing that RNAi targeting ZK686.3, *ribo-1*, *stt-3*, *ostb-1*, or *ostd-1* substantially decreased the induction of *T24B8.5p::GFP*, a target reporter of PMK-1, upon PA14 infection (Fig 5A and 5B). We also obtained consistent qRT-PCR results by measuring the mRNA levels of three selected PMK-1-regulated ATF-7 targets, T24B8.5, C17H12.8, and K08D8.5 (Fig 5C and 5E) [25]. In contrast, RNAi against each of the OST complex components did not affect the induction of *irg-1* or *irg-2* (Fig 5F and 5G), targets of another immune-regulatory transcription factor ZIP-2 [45]. These data suggest that the OST complex is required for the transcriptional responses caused by PA14 infection and subsequent PMK-1/p38 MAP kinase signaling.

## OST inhibition increases susceptibility to PA14 in a PMK-1-dependent manner

Next, we investigated the functional relevance of the link between the OST complex and UPR$^{ER}$ or PMK-1 signaling for immunity against PA14. We first measured the susceptibility of UPR$^{ER}$-impaired *xbp-1(-)* mutants treated with *stt-3* RNAi to PA14. We found that *stt-3* RNAi further reduced the survival of *xbp-1* mutants on PA14 (Fig 6A), suggesting that inhibition of the OST complex decreases the survival of worms on PA14, at least partially independently of UPR$^{ER}$. In addition, we found that PA14 infection slightly increased the level of *hsp-4p::GFP* in control worms but substantially decreased that in worms treated with *stt-3* RNAi (S5A Fig). Moreover, the induction of ER stress by tunicamycin significantly increased the survival of worms on PA14 (S5B Fig). These data suggest that elevated ER stress is not a major cause for the increased susceptibility to PA14 by the genetic inhibition of OST.

Next, we showed that *stt-3* RNAi completely suppressed the PA14 resistance conferred by the transgenic expression of *hsp-60* (Fig 6B), which we previously showed to up-regulate PMK-1 signaling [32]. Conversely, we found that decreased survival of worms on PA14 by *stt-3* RNAi was not additive to that by *pmk-1(-)* mutations (Fig 6C). Although RNAi knockdown has a limitation for genetic interaction analysis, these data are consistent with the possibility that OST and PMK-1 act in the same pathway to decrease the susceptibility of worms on PA14. These results are also consistent with our transcriptomic data showing that inhibition of the OST complex decreases PMK-1 signaling on PA14 infection.

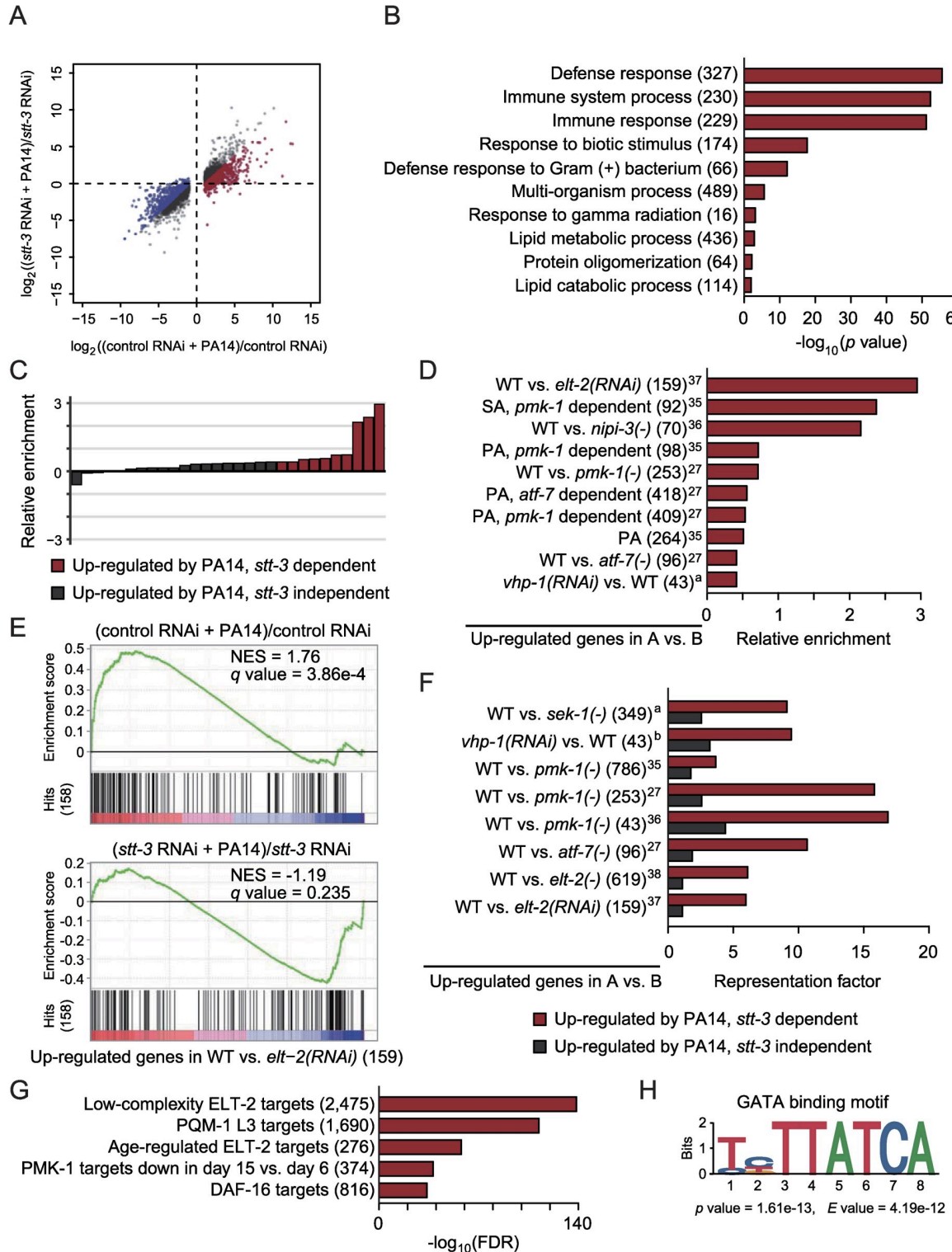

**Fig 4. OST complex mediates PMK-1-dependent induction of a subset of PA14-responsive genes.** (**A**) A scatter plot showing the effects of RNAi targeting *stt-3* on the expression levels of mRNAs that were significantly changed by PA14 infection (fold change > 2, BH-adjusted *p* value < 0.05). Red dots and blue dots indicate *stt-3*-dependent 569 up- and 569 down-regulated genes, respectively (fold change > 2 and BH-adjusted *p* value < 0.05 by PA14 in control RNAi, but the changes were reduced by fold > 2 by *stt-3* RNAi). (**B**) Overrepresented GO terms of genes induced by PA14 in an *stt-3*-dependent manner. The numbers of genes for individual terms are indicated in parentheses. (**C**) Twenty-nine gene sets whose expression was significantly increased by PA14 (*q* value < 0.25). The relative

enrichment (NES$_{(ctrl\,RNAi\,+\,PA14)/ctrl\,RNAi}$−NES$_{(stt\text{-}3\,RNAi\,+\,PA14)/stt\text{-}3\,RNAi}$) indicates the extent of expression changes of a gene set by PA14 with or without *stt-3* RNAi. Red bars represent 10 *stt-3*-dependent PA14-induced sets. Grey bars represent *stt-3*-independent PA14-induced sets, some of which are shown in S4 Fig. (**D**) *stt-3*-dependent PA14-enhanced gene sets. In addition to PMK-1 signaling component genes, we noticed that genes whose activation was dependent on NIPI-3, another regulator of immunity against PA14 [42], were associated with genes up-regulated by PA14 in an *stt-3*-dependent manner. PA: PA14. SA: *Staphylococcus aureus*. Superscript numbers indicate references that include corresponding transcriptomic data. [a] data (GSE82238) in GEO. The numbers of genes for individual categories are indicated in parentheses. (**E**) Genes up-regulated in an ELT-2-dependent manner [43] were induced by PA14 in an *stt-3*-dependent manner. NES: normalized enriched score. *q* values were obtained by calculating the false discovery rate (FDR) corresponding to each NES. (**F**) Representation factors for overlaps between *stt-3*-dependent or *stt-3*-independent PA14-induced genes and PMK-1 signaling component-encoding genes. [a] data (GSE92902) and [b] data (GSE82238) in GEO. The numbers of genes for individual categories are indicated in parentheses. (**G**) Targets of transcription factors specifically associated with *stt-3*-dependent PA14-induced genes. The numbers of genes for individual categories are indicated in parentheses. (**H**) A GATA binding motif (MA0542.1) in JASPAR CORE [81] enriched in promoter regions (-1 kb to +100 bp) of *stt-3*-dependent PA14-induced genes compared to corresponding positions of *stt-3*-independent PA14-induced genes.

We then determined the genetic interaction among *stt-3* RNAi, UPR$^{ER}$, and PMK-1 for defense against PA14. We first established that *xbp-1(-)* mutation does not further enhance PA14 susceptibility caused by a *pmk-1(-)* mutation (S6 Fig), consistent with a previous report [31]. Importantly, *stt-3* RNAi did not decrease the shortened survival of *xbp-1(-); pmk-1(-)* mutants upon PA14 infection (Fig 6D). Altogether, our data indicate that inhibition of the OST complex, which perturbs UPR$^{ER}$, increases susceptibility to PA14 through down-regulating PMK-1 signaling.

## Discussion

### The OST complex contributes to defense against pathogenic *P. aeruginosa* through affecting transcriptional responses to infection

In this study, we showed that the OST complex was required for the protection of *C. elegans* against pathogenic *P. aeruginosa*, PA14. We found that the OST complex mediated ER protein homeostasis through maintaining UPR$^{ER}$. Defects in the N-glycosylation process likely disrupt the maturation and the proper localization of ER proteins, thereby causing ER stress via the accumulation of abnormal proteins in the ER. Our transcriptomic analysis revealed that the OST complex contributed to transcriptional responses to PA14 infection. In particular, the OST complex was required for PMK-1-dependent gene induction in response to PA14 infection. We also showed that PMK-1 was required for OST-mediated defense against PA14. Overall, our work suggests that an evolutionarily conserved OST complex contributes to protection against pathogenic bacteria by maintaining ER homeostasis and by up-regulating PMK-1 signaling (Fig 6E).

### Substrate proteins of the OST complex may contribute to systemic immune response

Many secreted and membrane-localized proteins constitute the substrates of the OST complex whose stability, maturation, or localization is regulated by N-linked glycosylation. *C. elegans* possesses many anti-microbial effector proteins that are secreted from cells upon exposure to various pathogenic microbes [52, 53]. Thus, the OST complex may modulate immune responses by regulating the post-translational modification, intracellular trafficking, and/or secretion of immune effector proteins such as anti-microbial peptides, cytokines, and receptor proteins. VIT-6 is secreted by intestinal cells and transported into oocytes for lipid transfer from parents to offspring [38, 39, 54–56]. Our data suggest that OST mediates N-glycosylation of VIT-6 to maintain defense against PA14 infection. It remains undetermined how VIT-6 contributes to protection against PA14 infection. One possibility is that the transport of lipids

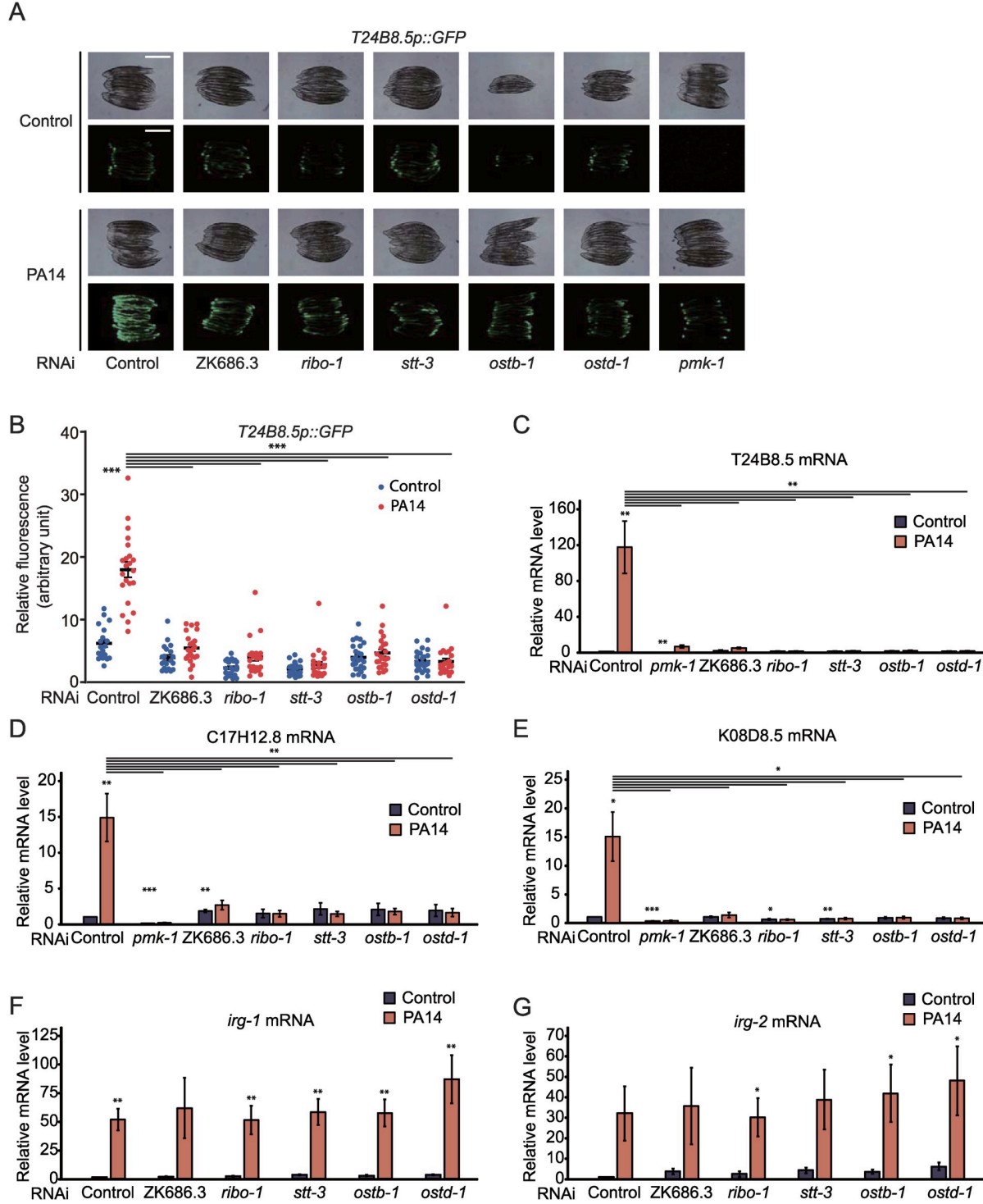

**Fig 5. OST complex increases the expression of several PMK-1-regulated genes under PA14-infected conditions.** (**A**) RNAi targeting each OST component (ZK686.3, *ribo-1*, *stt-3*, *ostb-1*, or *ostd-1*) reduced the induction of *T24B8.5p::GFP* by PA14 infection. *pmk-1* RNAi was used as a positive control. Scale bar indicates 500 μm. Different from the genetic inhibition of the OST complex, treatment with tunicamycin (5 μg/ml) did not alter the level of *T24B8.5p::GFP* with or without PA14 infection ([S5C Fig]). (**B**) Quantification of data in panel **A** (n ≥ 20 from three independent experiments). (**C-E**) qRT-PCR analysis data showing the mRNA levels of three selected PMK-1-regulated genes, T24B8.5 (**C**), C17H12.8 (**D**), and K08D8.5 (**E**), upon knocking down indicated OST components with or without PA14 infection (n ≥ 3). *pmk-1* RNAi was used as a positive control. (**F-G**) qRT-PCR analysis data showing the mRNA levels of two selected ZIP-2-regulated genes, *irg-1* (**F**) and *irg-2* (**G**), upon knocking down indicated OST components with or without PA14 infection (n = 4). Error bars indicate SEM. *p* values were calculated by using two-tailed Student's *t*-test (*$p < 0.05$, **$p < 0.01$, ***$p < 0.001$).

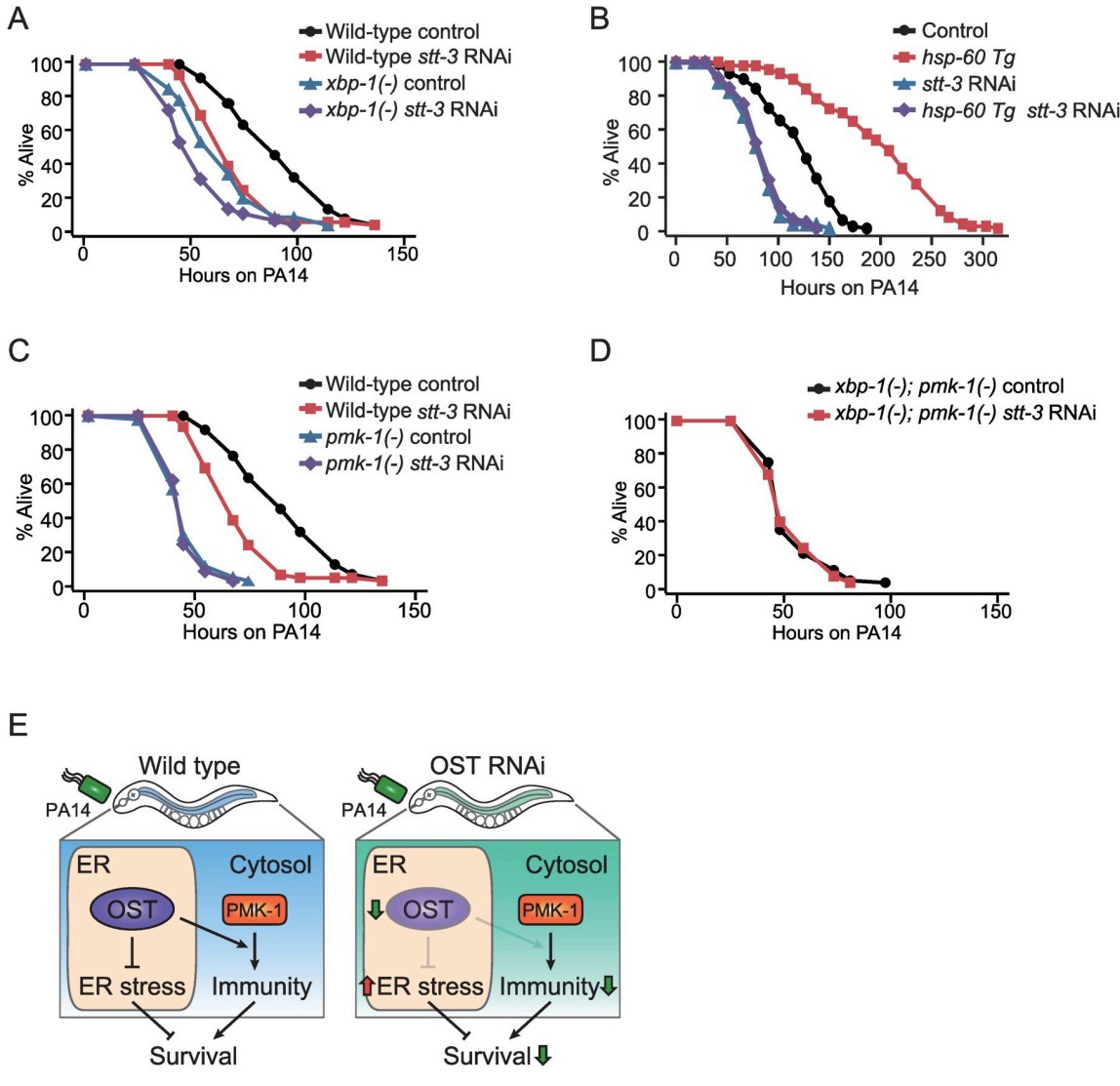

**Fig 6. OST complex reduces susceptibility to PA14 through PMK-1 signaling.** (**A**) Genetic inhibition of OST complex by *stt-3* RNAi further decreased the survival of *xbp-1(zc12)* [*xbp-1(-)*] animals on PA14. (**B-C**) *stt-3* RNAi suppressed the enhanced PA14 resistance of *hsp-60*-trangenic worms (*hsp-60 Tg*) (**B**), but did not reduce the survival of *pmk-1(km25)* [*pmk-1(-)*] mutants (**C**) upon PA14 infection. (**D**) *stt-3* RNAi did not affect the PA14 susceptibility of *xbp-1(-); pmk-1(-)* animals. We also obtained data suggesting that VIT-6 affects sensitivity to PA14 independently of HSP-60 and PMK-1 pathway (See S7 Fig. legends for details). (**E**) Schematic model. OST contributes to the maintenance of worm survival on PA14 infection by decreasing stress in the ER and by up-regulating PMK-1-mediated immunity in the cytosol. See S3 Table for additional repeats and statistical analysis for survival assay results in this figure.

from intestinal cells to oocytes may play a role in preventing PA14 pathogenicity. Alternatively, VIT-6 may function as a secreted factor in signal transduction for improving anti-bacterial immunity, in addition to or instead of its potential role in lipid transport. It will be interesting to test these possibilities in future studies.

## OST may contribute to PMK-1-dependent immunity via maintaining glycoprotein quality control in the ER

Previous studies with *C. elegans* have provided evidence for the crucial role of ER protein homeostasis in PMK-1-dependent immunity [29, 31]. Under PA14 infection conditions,

PMK-1 up-regulates many ER protein-coding genes, eventually increasing the protein load and the ER stress [29]. XBP-1-mediated UPR$^{ER}$ is a critical factor for immunity against PA14 (Fig 6A) [29, 31, 39]. Similar to XBP-1 and UPR$^{ER}$, OST-regulated N-glycosylation is crucial for ER protein homeostasis (Fig 3A–3H) [3, 57], and we showed that the OST contributes to protection against PA14 infection. Interestingly, our transcriptomic data indicate that PA14 infection down-regulated many *abu* genes (*abu-1*, *6*, *7*, *9*, *10*, and *15*) and *pqn* genes (*pqn-2*, *37*, *57*, *60*, *63*, and *74*), which were suppressed by the inhibition of OST (S5 Table). *abu* and *pqn* genes regulate non-canonical UPR$^{ER}$ [58] and also contribute to the protection of worms against pathogenic bacteria [59, 60]. Research that determines how OST regulates canonical and non-canonical UPR$^{ER}$ processes for influencing PMK-1 signaling will be crucial for elucidating N-glycosylation-mediated defense mechanisms against pathogens.

## Potential role of N-glycosylation and OST complex in mammalian immunity

Recent studies suggest that the mammalian OST complex regulates the N-glycosylation of immune regulators to influence immunity. A previous study using a CRISPR/Cas9 knockout screen identified multiple OST complex components as regulators of cytokine production and inflammatory responses in primary dendritic cells [61]. In addition, N-glycosylation of mammalian toll-like receptors appears to be critical for their function as immune regulators [62, 63]. Many cytokines are also N-glycosylated for proper secretion and/or activity [64]. Our current study using *C. elegans* suggests novel mechanisms for immune regulation by the OST, through N-glycosylation of defense factors, which transmit infection signals from the ER to the PMK-1 pathway in cytosol. It will be interesting to perform further studies investigating whether similar processes are conserved in mammals.

## Materials and methods

### *C. elegans* strains

All strains were maintained as previously described [65]. Some strains were obtained from Caenorhabditis Genetics Center, which is funded by the NIH National Center for Resources (p40 OD010440). Strains used in this study were described below: N2 wild-type, PHX458 *stt-3 (syb458)/hT2*, SJ4005 *zcIs4[hsp-4p::GFP]*, AU78 *agIs219[T24B8.5p::GFP]* a gift from D.H. Kim lab, IJ939 *yhIs62[hsp-60p::hsp-60::GFP; odr-1p::RFP]*, IJ940 *yhIs63[hsp-60p::hsp-60::GFP; odr-1p::RFP]*, IJ1918 *xbp-1(zc12)* obtained by outcrossing SJ17 four times to Lee-lab N2, IJ130 *pmk-1(km25)* obtained by outcrossing KU25 four times to Lee-lab N2, IJ1935 *xbp-1(zc12); pmk-1(km25)*, GA2100 *bIs1[vit-2p::vit-2::GFP + rol-6]; wuEx277[vit-6p::vit-6::mCherry + rol-6]* a gift from D. Gems lab. PHX458, STT-3 catalytic residue mutant (Y550A) strain was designed based on a report characterzing the yeast STT3 [33], and generated using CRISPR/Cas9 by SunyBiotech (http://www.sunybiotech.com/).

### Pathogen killing assays

PA14 standard (small lawn) slow killing assay was performed as previously described [32, 66]. Briefly, PA14 was cultured in LB media at 37˚C overnight, and 5 μl of the liquid culture was then seeded on the center of high peptone NGM plates (0.35% bactopeptone). The plates were incubated at 37˚C for 24 hrs and kept at 25˚C before use. Wild-type (N2) or mutant worms were grown on plates where RNAi bacteria were seeded with 1 mM IPTG (Isopropyl β-D-1-thiogalactopyranoside). L4 or young adult stage worms were transferred to the PA14-seeded plates with 50 μM FUdR (5-fluoro-20-deoxyuridine) that prevents progeny from hatching.

The worms were maintained at 25°C, scored twice a day and counted as dead if the worms did not respond to prodding. For PA14 big lawn killing assays, 15 μl of PA14 liquid culture was seeded onto each high peptone NGM plate and subsequently spread out through entire surface of the plate. PA14 fast killing assay was performed as previously described with minor modifications [67]. Briefly, 15 μl of PA14 liquid culture was spread on PGS (peptone-glucose-sorbitol) plates, incubated for 24 hrs at 37°C, and kept at 25°C for 2 hrs before use. L4 stage larval wild-type worms that were pre-treated with control or *stt-3* RNAi were transferred to PA14 on PGS for survival assays. Pathogen killing assays against *E. faecalis* or pathogenic *E. coli* were performed as previously described [17, 68]. Statistical analysis of survival data was performed by using OASIS (http://sbi.postech.ac.kr/oasis) and OASIS2 (https://sbi.postech.ac.kr/oasis2), which calculate *p* values using log-rank (Mantel-Cox method) test [69, 70].

## Measurement of pharyngeal pumping rate

Wild-type N2 worms were grown on control or *stt-3* RNAi plates. The pumping rate of young adult stage worms was counted under a dissecting microscope for 15 sec and then re-scaled to pumping/1 min.

## Intestinal PA14-GFP accumulation assays

Intestinal PA14-GFP accumulation assays were performed as previously described [22, 32], with minor modifications. L4 stage larval worms that were pre-treated with control RNAi or RNAi targeting each OST component were infected with PA14 that expresses GFP (PA14-GFP) under slow killing and small lawn conditions for 72 hrs and subsequently imaged using a microscope.

## Microscopy

Transgenic animals that expressed fluorescent proteins or wild-type animals infected with PA14-GFP were imaged by using AxioCam HRc (Zeiss Corporation, Jena, Germany) camera mounted on a Zeiss Axioscope A.1 microscope (Zeiss Corporation, Jena, Germany) upon treating with 100 mM sodium azide (DAEJUNG, Siheung, South Korea) for anesthesia. The fluorescence intensity of the animals was quantified by using ImageJ software (Rasband, W.S., ImageJ, U.S. National Institute of Health, Bethesda, Maryland, USA, http://rsb.info.nih.gov/ij/). The quantification of the data was displayed by using Graphpad Prism 7 software (http://www.graphpad.com/scientific-software/prism/).

## Identification of PA14 infection-dependent glycoproteins

Eggs of wild-type N2 worms were placed on control RNAi or *stt-3* RNAi bacteria-seeded plates and grown to L4 stage. L4 stage worms were then transferred to corresponding RNAi bacteria- or PA14-seeded plates after washing with M9 buffer 3–4 times. After 12 hrs, worms were harvested after washing with M9 buffer 3–4 times to remove residual bacteria and immediately frozen with liquid nitrogen. The worm samples were ground to form frozen powders with a cold mortar and pestle using liquid nitrogen. The powders dissolved in IPG dry strips (4–10 NL IPG, 24 cm, Genomine, Korea) were equilibrated for 12–16 hrs with 7 M urea, 2 M thiourea containing 2% 3-[(3-cholamidopropy)dimethyammonio]-1-propanesulfonate (CHAPS), 1% dithiothreitol (DTT), 1% pharmalyte and respectively loaded with 200 μg of the sample. Isoelectric focusing (IEF) was performed at 20°C using a Multiphor II electrophoresis unit and EPS 3500 XL power supply (Amersham Biosciences) following manufacturer's instruction. For IEF, the voltage was linearly increased from 150 to 3,500 V during 3 hrs for sample entry

followed by constant 3,500 V and focusing was completed after reaching 96 kVh. Prior to the second dimensional separation, strips were incubated for 10 min in equilibration buffer (50 mM Tris-Cl, pH 6.8 containing 6 M urea, 2% SDS and 30% glycerol), firstly with 1% DTT and secondly with 2.5% iodoacetamide. Equilibrated strips were inserted onto SDS-PAGE gels (20 x 24 cm, 10–16%). SDS-PAGE was performed using Hoefer DALT 2D system (Amersham Biosciences) following manufacturer's instruction. The 2D gels were run at 20°C for 1,700 Vh. Subsequently, the protein gels were stained with Pro-Q™ Emerald 488 glycoprotein stain kit (Invitrogen) and imaged for measuring glycoprotein levels. The gels were then stained by using a Colloidal Coomassie Brilliant Blue (CBB) method for measuring the levels of total proteins. Quantitative analysis of digitized images was carried out using the PDQuest (version 7.0, BioRad) software following the protocol provided by the manufacturer. The intensities of glycoproteins were normalized to those of total proteins, to compare the glycoprotein levels in four different experimental samples. Protein spots whose glycosylation levels were increased by PA14 infection in an *stt-3*-dependent manner were selected for extracting proteins from the gels and for subsequent Peptide Mass Fingerprinting (PMF) analyses.

## Lifespan assays

Lifespan assays were performed as previously described [71]. Young adult worms that were grown on control RNAi and OST component RNAi plates were transferred to 5 μM FUdR-containing control and OST component RNAi plates, respectively. The animals that did not move upon gentle touch were counted as dead. The animals that crawled off the plates, displayed ruptured vulvae, burrowed, or displayed internal hatching were censored but included in the subsequent statistical analysis. Statistical analysis of lifespan data was performed by using OASIS (http://sbi.postech.ac.kr/oasis), and OASIS2 (https://sbi.postech.ac.kr/oasis2), which calculate *p* values using log-rank (Mantel-Cox method) test [69, 70].

## RNA extraction and quantitative RT-PCR

RNA extraction and qRT-PCR were performed as previously described with minor modifications [72]. Wild-type N2 worms fed with control RNAi, *pmk-1* RNAi or OST component RNAi bacteria were grown to L4 stage and then transferred to corresponding RNAi bacteria- or PA14-seeded plates after washing with M9 buffer 3–4 times. After 12 hrs, worms were harvested by washing with M9 buffer 3–4 times to remove residual bacteria. Total RNA in the animals was isolated by using RNAiso plus (Takara, Seta, Kyoto, Japan) and was used for synthesizing cDNA with ImProm-II Reverse Transcriptase kit (Promega, Madison, WI, USA). Quantitative real-time PCR was performed by using StepOne Real-Time PCR System (Applied Biosystems, Foster City, CA, USA) as described in the manufacturer's protocol. Comparative $C_T$ method was used for the quantitative analysis of mRNAs. For all biological data sets, *ama-1* mRNA, which encodes an RNA polymerase II large subunit, was used as a normalization control. The primer sequences used for qRT-PCR analysis are as follows; 5'-TGGAACTCTGGAGTCACACC (forward) and 5'-CATCCTCCTTCATTGAACGG (reverse) for *ama-1*, 5'-AGTTGAAATCATCGCCAACG (forward) and 5'- GCCCAATCAGACGCTTGG (reverse) for *hsp-4*, 5'- CCGATCCACCTCCATCAAC (forward) and 5'-ACCGTCTGCTCCTTCCTCAATG (reverse) for total *xbp-1*, 5'-TGCCTTTGAATCAGCAGTGG (forward) and 5'- ACCGTCTGCTCCTTCCTAATG (reverse) for spliced *xbp-1*, 5'-TGTTAGACAATGCCATGATGAA (forward) and 5'-ATTGGCTGTGCAGTTGTACC (reverse) for T24B8.5, 5'-GAACAATAGTGTCAAGCCGATCTGC (forward) and 5'-TTCTGAATGATGAATGCATGTTTAC (reverse) for C17H12.8, 5'-TCTGGTCAAAATATCCTCCGGGAAG (forward) and 5'-GAGCATCACTCGATTGATTGCAGTG (reverse) for K08D8.5, 5'-GATCTTGTTCCGTACCCATGG (forward)

and 5'-GCTTTGTCAAGACCAATTCCC (reverse) for *irg-1*, 5'- TTTACTTCCGAAAATCTC TC (forward) and 5'-GATAAGTTTTGACAATTGTG (reverse) for *irg-2*.

## RNA-sequencing analysis

RNA was extracted as described above (see RNA extraction and quantitative RT-PCR section). Libraries were constructed and sequencing was performed by Macrogen Inc. (Seoul, South Korea). Sequencing reads were aligned to the *C. elegans* genome WBcel235 (ce11) by using HISAT2 (v.2.0.5) [73]. Aligned transcripts were assembled and quantified by using StringTie (v.1.3.3b) [74] and Ballgown (v.2.0.0) [75]. The batch effects of samples were removed by upper-quartile normalization followed by RUVSeq with control genes (v1.16.1) [76]. Differentially expressed genes (fold change > 2 and adjusted *p* value < 0.05) were identified by using DESeq2 (v.1.22.2) [77]. Wald test *p* values are adjusted for multiple testing using the procedure of Benjamini and Hochberg. Genes whose expression was significantly changed by PA14 in control RNAi-treated worms but not by PA14 in *stt-3* RNAi-treated worms (fold change > 2 in opposite directions) were defined as *stt-3*-dependent PA14-responsive genes (569 genes for *stt-3*-dependent up-regulated genes by PA14 and 569 genes for *stt-3*-dependent down-regulated genes by PA14, S5 Table); it is a rare coincidence for having exact 569 genes for each set. GO terms enriched in certain genes were identified by GOstats (v.2.48.0) [78]. The ratio of spliced form in a particular transcript was obtained by using MISO (v.0.5.4) [79]. Motifs enriched in the promoter regions (-1 kb to +100 bp) of genes of interest compared to corresponding positions of background genes were identified by using AME (v.5.0.5) [80]. Given gene lists with previously discovered targets of transcription factors were compared by using WormExp (v.1.0) [40, 81]. Fifty-two previously published gene sets related to anti-PA14 immunity were collected. Global changes of certain gene sets in a comparison (treatment vs. control) were shown as normalized enrichment scores (NES) by using GSEA (v.3.0) [82]. A difference between two comparisons was defined as a relative enrichment: $NES_{(ctrl\ RNAi\ +\ PA14)/ctrl\ RNAi} - NES_{(stt-3\ RNAi\ +\ PA14)/stt-3\ RNAi}$. The numbers represent the differences in expression changes caused by PA14 treatment with or without *stt-3* RNAi for specific gene sets. To identify gene sets significantly changed by PA14, gene sets whose FDR *q* value < 0.25 in any comparison were selected. R (v.3.5.3, http://www.r-project.org) was used for plotting results. Raw data and processed data are available in Gene Expression Omnibus (https://www.ncbi.nlm.nih.gov/geo, GSE134687).

## Supporting information

**S1 Fig. The effects of *stt-3* RNAi on the survival of worms infected with PA14 at L4 and young adult stages and pumping rates, and the schematic of *stt-3*(*syb458*) knock-in allele.** (**A-B**) *stt-3* RNAi reduced the survival of worms infected with PA14 at the L4 (**A**) and young adult (**B**) stages. *pmk-1* RNAi was used as a positive control. (**C**) The molecular nature of *syb458* knock in mutation in *stt-3*. Upper part: the gene structure of *C. elegans stt-3*. The location of *stt-3 (syb458)* mutation was marked with a red inverted triangle. Bottom part: the comparison of worm STT-3, yeast STT3, mouse STT3a/b and human STT3A/B protein sequences that contain conserved WWDYG motifs (marked with a black line). Black bars at the bottom indicate conservation scores of individual amino acid residues. WWDYG: Trp-Trp-Asp-Tyr-Gly. *stt-3(syb458)* mutation results in the substitution of 550th tyrosine to alanine that perturbs the WWDYG motif. (**D**) Pharyngeal pumping rates of worms treated with control or *stt-3* RNAi. Error bars indicate SEM. *p* values were calculated by using two-tailed Student's *t*-test (**$p$ < 0.01). See S3 Table for additional repeats and statistical analysis for survival assay results in this figure. (EPS)

**S2 Fig. Enriched GO terms and *xbp-1* splicing affected by *stt-3* RNAi.** (**A**) Overrepresented GO terms of genes whose expression was decreased by *stt-3* RNAi. The numbers of genes for individual terms are indicated in parentheses. (**B**) qRT-PCR data showing that feeding worms with *stt-3* RNAi bacteria was sufficient to reduce *stt-3* mRNA levels. (**C-D**) qRT-PCR data showing that increases in the mRNA levels of *hsp-4* (**C**) and *hsp-3* (**D**) by *stt-3* RNAi were suppressed by *xbp-1* mutations. (**E**) An intron retention of *xbp-1* transcripts in two other sets of RNA-seq data, which are different from Fig 3G. A grey box with dotted lines indicates the location of alternative splicing. Error bars indicate SEM. *p* values were calculated by using two-tailed Student's *t*-test (*$p < 0.05$, **$p < 0.01$, ***$p < 0.001$).
(EPS)

**S3 Fig. Infection with PA14 causes broad remodeling of transcription.** (**A**) A principal component (PC) analysis showing relative distance between samples. (**B**) Overrepresented GO terms of genes down-regulated by PA14 in an *stt-3*-dependent manner. UPR$^{ER}$: ER unfolded protein response. The numbers of genes for individual terms are indicated in parentheses.
(EPS)

**S4 Fig. Relationship between *stt-3*-independent PA14-induced genes and other immune-regulators.** A subset of genes whose expression was increased by PA14 in an *stt-3*-independent manner, shown in Fig 4C. The relative enrichment indicates the extent of expression changes in a gene set by PA14 infection with or without *stt-3* RNAi. The numbers of genes for individual categories are indicated in parentheses. Superscript numbers indicate supplemental references that include corresponding transcriptomic data.
(EPS)

**S5 Fig. ER stress does not appear to be a main cause for the susceptibility of OST-inhibited worms to PA14.** (**A**) Fluorescent images of *hsp-4p::GFP* worms. PA14 infection slightly increased the expression level of *hsp-4p::GFP* in control worms but substantially decreased that in worms treated with *stt-3* RNAi. Scale bar indicates 500 μm. (**B**) Tunicamycin significantly increased the survival of wild-type (WT) worms on PA14. (**C**) Fluorescent images showing the effect of ER stress induced by treatment with tunicamycin on the induction of *T24B8.5p::GFP* reporter with or without PA14 infection. These data suggest that down-regulation of PMK-1 pathway by inhibition of OST is uncoupled by the induction of ER stress. Scale bar indicates 100 μm. See S3 Table for the additional repeats and statistical analysis for survival assay data in this figure.
(EPS)

**S6 Fig. PA14 susceptibility of animals with reduced UPR$^{ER}$ and PMK-1 signaling.** Consistent with a precious report [29], *xbp-1(zc12)* [*xbp-1(-)*] mutations did not further reduce the survival of *pmk-1(km25)* [*pmk-1(-)*] mutants on PA14. See S3 Table for additional repeats and statistical analysis for survival assay results in this figure.
(EPS)

**S7 Fig. VIT-6 does not appear to affect PMK-1 pathway.** (**A**) Genetic inhibition of *vit-6* by RNAi significantly reduced the survival of both wild-type (WT) worms and *hsp-60* transgenic (*hsp-60 Tg*) worms to a similar extent. (**B**) *vit-6* RNAi did not alter the expression level of *T24B8.5p::GFP* reporter with or without PA14 infection. Scale bar indicates 500 μm. See S3 Table for the additional repeats and statistical analysis for survival assay data in this figure.
(EPS)

**S1 Table. Genetic screen data using PA14 standard slow killing assay with RNAi clones targeting non-mitochondrial genes that have mitochondrial GO terms.** S1 Table is uploaded as

a separate Excel file.
(XLSX)

**S2 Table. List of the proteins whose N-glycosylation levels were increased by PA14 infection in an *stt-3*-dependent manner.** S2 Table is uploaded as a separate word file.
(DOCX)

**S3 Table. The effects of genetic inhibition of each oligosaccharyl transferase (OST) component or *vit-6* on survival in various genetic backgrounds under pathogenic bacteria- or dietary *E. coli*-fed conditions.** S3 Table is uploaded as a separate word file.
(DOCX)

**S4 Table. Genes whose expression was significantly changed by *stt-3* RNAi in RNA-seq data.** S4 Table is uploaded as a separate Excel file.
(XLSX)

**S5 Table. Genes whose expression was significantly changed by PA14 in an *stt-3*-dependent manner.** S5 Table is uploaded as a separate Excel file.
(XLSX)

## Acknowledgments

We thank Drs. Dennis H. Kim and David Gems for providing some *C. elegans* strains. We also thank Lee lab members for helpful discussions and valuable comments on the manuscript.

## Author Contributions

**Conceptualization:** Dae-Eun Jeong, Joo-Yeon Yoo, Tae-Young Roh, Seung-Jae V. Lee.

**Data curation:** Dae-Eun Jeong, Yujin Lee, Seokjin Ham.

**Formal analysis:** Dae-Eun Jeong.

**Funding acquisition:** Joo-Yeon Yoo, Seung-Jae V. Lee.

**Investigation:** Dae-Eun Jeong, Yujin Lee, Seokjin Ham, Dongyeop Lee, Sujeong Kwon, Hae-Eun H. Park.

**Methodology:** Dae-Eun Jeong, Sun-Young Hwang.

**Project administration:** Seung-Jae V. Lee.

**Resources:** Dae-Eun Jeong, Seung-Jae V. Lee.

**Software:** Dae-Eun Jeong.

**Supervision:** Seung-Jae V. Lee.

**Visualization:** Dae-Eun Jeong, Yujin Lee, Seokjin Ham.

**Writing – original draft:** Dae-Eun Jeong, Yujin Lee, Seokjin Ham, Seung-Jae V. Lee.

**Writing – review & editing:** Seung-Jae V. Lee.

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
