## [Decision Letter · Decision Letter 0]

5 Sep 2019

Dear Dr Lee,

Thank you very much for submitting your Research Article entitled 'Oligosaccharyl transferase that maintains ER homeostasis enhances immunity via up-regulating p38 MAP kinase signaling in C. elegans' to PLOS Genetics. Your manuscript was fully evaluated at the editorial level and by independent peer reviewers. The reviewers appreciated the attention to an important problem, but raised some substantial concerns about the current manuscript. Based on the reviews, we will not be able to accept this version of the manuscript, but we would be willing to review again a much-revised version. We cannot, of course, promise publication at that time.

If you decide to revise the manuscript for further consideration at PLOS Genetics, please aim to resubmit within the next 60 days, unless it will take extra time to address the concerns of the reviewers, in which case we would appreciate an expected resubmission date by email to plosgenetics@plos.org.

[LINK]

We are sorry that we cannot be more positive about your manuscript at this stage. Please do not hesitate to contact us if you have any concerns or questions.

Yours sincerely,

Danielle A. Garsin

Associate Editor

PLOS Genetics

Gregory P. Copenhaver

Editor-in-Chief

PLOS Genetics

Reviewer's Responses to Questions

**Comments to the Authors:**

Reviewer #1: The authors present the argument that OST is involved in innate immunity and that it acts by regulating PMK-1 signaling. The biggest problem that I have with this argument is that the authors never conclusively prove that OST directly regulates innate immunity, at least not as it is conventionally considered.

Infection with PA14 causes ER stress by activation of the PMK-1 pathway. This is well established through the work of Dennis Kim. This activates the IRE-1/XBP-1 ER stress axis, as they note in their manuscript. Activation of this pathway stimulates the expression not only of chaperones, but also changes the expression of nuclear genes to reduce total gene expression to reduce the stress on the system. This is consistent with what has been reported in many model systems, and with what has been shown by the Kim lab in the context of PA14 infection.

But disrupting the OST system also induces ER stress, because of ER protein folding control pathways that utilize ERAD. This is not surprising; treatment of cells with tunicamycin induces ERAD and disrupting ERAD pathways and then treating with tunicamycin causes cell death, probably by overloading ER stress management pathways and inducing apoptosis, perhaps through the IRE1/XBP axis or through PERK. Indeed, it is known that members of the OST complex, when disrupted, have this exact effect. Given the absence of somatic apoptosis in C. elegans, it may not be surprising that the host succumbs to this stress, as was seen previously in Dr. Kim's work.

Instead, I think that it is more likely that the ER stress induced is simply overwhelming host stress responses and killing the worm. Admittedly, innate immunity and stress responses almost certainly have a greater relationship and overlap than their categorical considerations generally allow. Nevertheless, calling this system innate immunity, without further evidence, seems unwarranted. If the authors want to say that this is innate immunity, rather than ER stress, they should show that the effects that they've observed are specific to pathogens. (Point of fact, the observation that it doesn't seem to happen with EPEC or with E. faecalis actually undermines their claim that it is innate immunity. While I understand that innate immune processes differ, there are also consequences of infection that are not necessarily resolved by or defended against by innate immunity.) For example, the authors could show that disruption of the OST complex does not change the response to tunicamycin or some other ER stress phenotype. Alternatively, the authors could try dirupting the ERAD system via RNAi of cup-2 or hrdl-1. In any case, it isn't accurate to say that the genes are induced by OST, since this is not a transcriptional regulator. Instead, it is more accurate to say that full induction requires OST.

I hurry to note that this does nothing to make the paper less interesting. Ultimately, whether it is innate immunity or stress response will hopefully become a semantic issue, but for the time being matters are considered separate, and so accuracy is encouraged.

On a smaller scale the authors make several other claims that I have problems with. For example, they claim that VIT-6 glycosylation is increased by PA14 infection. There is no evidence presented to support this. They present an argument that VIT-6 glycosylation was decreased by stt-3 RNAi, which is not surprising, but there is insufficient information to understand how they came to that conclusion in the Materials and Methods. At the least, I missed the comparison between normal glycosylation of VIT-6 and post-infection glycosylation of VIT-6, but I think that it is absent from Supplementary Table 2.

Their slow kill data, while all consistent internally, don't make sense to me. Those assays are usually much faster. For example, 50% N2 killing at 4 days is really strange, often by that time, most of the worms are dead, especially if they are using L4 worms, which are more sensitive. For example, their colonization data are really strange. WT C. elegans by 72 h should be completely colonized by PA14 (e.g., Tan, et al. 1999; Irazoqui, et al., 2010; Twumasi-Boateng and Shapira, 2012; Kirienko, et al., 2013, etc.)

The authors should consider preparing a model figure to more graphically show their hypothesis about how this system works. They should also consider the alternative explanation. I.e., that overload of stress in the ER triggers retrograde signaling that reduces the expression of ER-targeted (or all) proteins, such as PMK-1 effectors.

Minor Issues:

VIT-6 should be hyphenated whenever it appears.

Please show a larger version of Fig 2A, perhaps even a part of the worm. It is difficult to see details in these images. This is also true for other fluorescence images, such as Fig 1C, Fig 5A, etc. It can be hard to tell correct expression from autofluorescence of dead worms, for example.

Were there 569 genes used for each anaylsis of stt-3? Was that done so that there were equal numbers in each case? I'm mostly curious, I don't think that it is a problem.

Reviewer #2: Jeong et al present a very interesting study looking at the OST protein complex on PA14 interactions and survival. This study is a nice complement from previous work from this group that investigated the role for mitochondrial function. The data is impressive and the study for the most part is well designed and I believe this work is an important contribution to the field. However, there are several issues in the presentation of the data and in several instances concerns with data interpretation and absence of followup experiments (or clarity of data). Overall I believe this work once revised will be a important to the field and readership of PLoS Gen.

Major concerns:

The authors demonstrate that loss of the stt-3 complex substantially decreased the resistance of C. elegans to PA14. But how do you reconcile potential lethality since loss of stt-3 is lethal.

More details are needed on how PA14 assays are performed since it has been demonstrated that L4 versus YA are very different responses and fast kill and slow kill responses are also different. In general, more experimental details required.

PA14 infects hosts and through secreted factor compromises health. The authors should look at potential impact of secreted factors on host health by comparing "fast-kill" and "slow-kill" assays.

The authors show that RNAi leads to accumulation of GFP labeled PA14, this should be normalized to GFP-labeled OP50 to account for RNAi impact on grinder function and bacterial clearance.

The impact of OST complex on VIT-6 is interesting, but this finding seems unfinished. Are the authors suggesting that the enhanced sensitivity is explained solely by VIT-6?

The authors use RNAi in combination with genetic mutants to infer epistasis. This is really not possible. The suggestion of parallel pathways cannot be established by these means. The authors need to use the stt-3 mutant in combination with pmk-1 mutants.

The discussion section is largely a restatement of the results. They authors should place their findings in the context of the field.

Minor comments:

The authors in several places use the word resistance when they should say sensitivity. N2 worms are never "resistant" to PA14. They eventually die, the question is how fast.

Figure 3C - many of these RNAi treatments have the potential to alter proteostasis. The authors should show RNA levels for hsp-4 also.

All Figures - the authors should define the number of genes that change in each category for enrichment.

Reviewer #3: The paper by Jeong et al provides an interesting outside-of-the-box examination of C. elegans immune responses. It demonstrates suppression of p38-dependent immune responses by disruption of ER homeostasis and ER stress. The particular driver of ER stress in this case is disruption of the oligosaccharyl transferase (OST) complex, which is required for protein glycosylation, reminiscent of the effects of the glycosylation inhibitor and popular ER stress agent tunicamycin. The authors show that genetic disruption of the OST complex affects glycosylation of three identified proteins, including VIT-6. Following OST disruption, VIT-6 becomes mislocalized, and further shows decreased induction upon P. aeruginosa strain PA14 infection. VIT-6 knock-down is sufficient to cause a significant decrease in resistance to this infection.

The results are intriguing, but leave a few questions unanswered, some of which could be addressed with simple experiments. These, as well as a few additional comments are described below:

1. Would exposure to tunicamycin also repress PA14 immune responses? This could easily be answered with the T24B8.5 reporter.

2. The authors suggest that VIT-6 may regulate immune responses. Would its knock-down be sufficient to abolish the protection conferred by the hsp-60 transgene-dependent activation of PMK-1?

3. Fig. 2A would benefit from better images. Also, what is the age of the presented worms? And what is the nature of VIT-6 milocalization? (from where to where)

Minor points:

1.“mutations in genes that act in the N-glycosylation pathway cause defects in both the innate and adaptive immune systems.” Add references.

2. Table S5 is not clear – Seems like the baseMean.ctrl_PA14_vs_ctrl_ctrl and baseMean.stt.3_PA14_vs_stt.3_ctrl columns have the same values. Better titles could help.

3. PA14 downregulated stt-3-dependent genes include some that were shown to be induced as part of a UPR in xbp-1 mutants (abu, pqn) (Urano J Cell Sci 2002), and were shown to be required for resistance to Salmonella (Haskins Plos Genet 2008). This seems to be relevant for the paper, and the authors may want to consider this.

4. Also, for the relationship between PMK-1, ATF-7 and ELT-2, the authors may want to consider Block Plos Genet 2015.

5. In discussion the authors write: ” In particular, many genes up-regulated in a PMK-1-dependent manner were induced by the OST complex upon PA14 infection.” I am not sure that this is the most accurate way to put this. How about “the OST complex is required for PMK-1-dependent gene induction in response to PA14 infection”? In the same vein, the title could be changed to: “Compromised ER homeostasis in C. elegans due to disruption of the Oligosaccharyl transferase complex suppresses p38-dependent immunity “

**Have all data underlying the figures and results presented in the manuscript been provided?**

Reviewer #1: Yes

Reviewer #2: Yes

Reviewer #3: Yes

PLOS authors have the option to publish the peer review history of their article (what does this mean?). If published, this will include your full peer review and any attached files.

Reviewer #1: No

Reviewer #2: No

Reviewer #3: No

---

## [Decision Letter · Decision Letter 1]

21 Dec 2019

Dear Dr Lee,

Thank you very much for submitting your Research Article entitled 'Compromised ER homeostasis by inhibition of the oligosaccharyl transferase in C. elegans suppresses p38-dependent protection against pathogenic bacteria' to PLOS Genetics. Your manuscript was fully evaluated at the editorial level and by independent peer reviewers. The reviewers appreciated the attention to an important topic, but one reviewer identified some aspects of the manuscript that he/she thought could be further improved and we encourage you to consider his/her comments.

We therefore ask you to modify the manuscript according to the review recommendations before we can consider your manuscript for acceptance. Your revisions should address the specific points made by each reviewer.

[LINK]

Yours sincerely,

Danielle A. Garsin

Associate Editor

PLOS Genetics

Gregory P. Copenhaver

Editor-in-Chief

PLOS Genetics

Reviewer's Responses to Questions

**Comments to the Authors:**

Reviewer #1: The authors have addressed all of my concerns. The manuscript is very interesting, and I can't wait to see where it goes next.

Reviewer #2: The authors have addressed my concerns

Reviewer #3: The authors have addressed my earlier comments satisfactorily. I feel that the paper is now clearer and stronger. However, the new experiments provide new insights that should be reflected in the title. In particular, results showing that tunicamycin did not compromise resistance to PA14 (Fig. S5B), together with the ability of stt-3 RNAi to further reduce PA14 resistance in xbp-1 mutants (Fig. 6), suggest that compromised ER proteostasis is not the primary cause of reduced PA14 susceptibility, and that something else associated with OST is the culprit. This should be reflected in the title, which currently ties ER proteostasis disruption to reduced infection resistance, while the new results suggest that the two phenotypes might be distinct consequences of OST disruption.

Further regarding the results in Fig. 5, saying that “ having established that the OST complex mediates the transcriptional changes caused by UPR-ER and PMK-1 signaling” (p.11, line 9) is somewhat misleading. The current phrasing implies that OST is part of the normal mechanism that regulates UPR-ER and innate immune responses, which does not seem to be the case. The phrasing may be almost true for PMK-1 signaling (although I would say that OST is required for PMK-1 signaling , rather than mediating transcription), but for UPR-ER, it is quite different, since OST disruption is what induces the UPR-ER.

Also, saying that “xbp-1 mutations further reduced the survival of stt-3 RNAi-treated worms on PA14 (Fig 6A)” is inaccurate, since RNAi does not achieve full knock out. The important results in this panel, which supports the authors’ conclusion is that stt-3 RNAi can further reduce resistance to PA14 of xbp-1 mutants.

**Have all data underlying the figures and results presented in the manuscript been provided?**

Reviewer #1: Yes

Reviewer #2: None

Reviewer #3: Yes

PLOS authors have the option to publish the peer review history of their article (what does this mean?). If published, this will include your full peer review and any attached files.

Reviewer #1: No

Reviewer #2: No

Reviewer #3: No

---

## [Editor Report · Decision Letter 2]

20 Jan 2020

Dear Dr Lee,

We are pleased to inform you that your manuscript entitled "Inhibition of the oligosaccharyl transferase in C. elegans that compromises ER proteostasis suppresses p38-dependent protection against pathogenic bacteria" has been editorially accepted for publication in PLOS Genetics. Congratulations!

Yours sincerely,

Danielle A. Garsin

Associate Editor

PLOS Genetics

Gregory P. Copenhaver

Editor-in-Chief

PLOS Genetics

Comments from the reviewers (if applicable):

**Data Deposition**

http://datadryad.org/submit?journalID=pgenetics&manu=PGENETICS-D-19-01292R2

**Press Queries**

---

## [Editor Report · Acceptance letter]

5 Feb 2020

PGENETICS-D-19-01292R2 

Inhibition of the oligosaccharyl transferase in Caenorhabditis elegan that compromises ER proteostasis suppresses p38-dependent protection against pathogenic bacteria 

Dear Dr Lee, 

We are pleased to inform you that your manuscript entitled "Inhibition of the oligosaccharyl transferase in Caenorhabditis elegan that compromises ER proteostasis suppresses p38-dependent protection against pathogenic bacteria" has been formally accepted for publication in PLOS Genetics! Your manuscript is now with our production department and you will be notified of the publication date in due course.

With kind regards,

Matt Lyles

PLOS Genetics

On behalf of:
